# Alleviating oxygen evolution from Li-excess oxide materials through theory-guided surface protection

Yongwoo Shin[1,4], Wang Hay Kan [1,5], Muratahan Aykol[1,6], Joseph K. Papp [2], Bryan D. McCloskey[1,2], Guoying Chen[1] & Kristin A. Persson [1,3]

Li-excess cathodes comprise one of the most promising avenues for increasing the energy density of current Li-ion technology. However, the first-cycle surface oxygen release in these materials causes cation densification and structural reconstruction of the surface region, leading to encumbered ionic transport and increased impedance. In this work, we use the first principles Density Functional Theory to systematically screen for optimal cation dopants to improve oxygen-retention at the surface. The initial dopant set includes all transition metal, post-transition metal, and metalloid elements. Our screening identifies Os, Sb, Ru, Ir, or Ta as high-ranking dopants considering the combined criteria, and rationalization based on the electronic structure of the top candidates are presented. To validate the theoretical screening, a Ta-doped $Li_{1.3}Nb_{0.3}Mn_{0.4}O_2$ cathode was synthesized and shown to present initial improved electrochemical performance as well as significantly reduced oxygen evolution, as compared with the pristine, un-doped, system.

[1] Energy Storage and Distributed Resources Division, Lawrence Berkeley National Laboratory, Berkeley, CA 94720, USA. [2] Department of Chemical and Biomolecular Engineering, University of California, Berkeley, CA 94720, USA. [3] Department of Materials Science and Engineering, University of California, Berkeley, Berkeley, CA 94704, USA. [4] Present address: Advanced Materials Lab, Samsung Research America, Burlington, MA 01803, USA. [5] Present address: Institute of High Energy Physics, Chinese Academy of Sciences, Guangdong 523803, PR China. [6] Present address: Toyota Research Institute, Los Altos, CA 94022, USA. Correspondence and requests for materials should be addressed to K.A.P. (email: kapersson@lbl.gov)

Structural transformation and chemical evolution at the interface of materials result from the interplay between a material and its environment during synthesis as well as under operating conditions. The resulting surface structure can impact the performance of the material[1,2] and hence is relevant to a wide variety of applications spanning energy storage, catalysis, alloy design, etc. In rechargeable Li-ion energy storage, the structural and chemical rearrangement of the electrode surface during the charge–discharge process is a possible culprit for detrimental passivation of the active material. In particular, the class of high-energy density, Li-excess cathode materials, which has shown promise as the next generation Li-ion battery technology[2,3], are prone to extensive oxygen loss[2,4,5] and the resulting formation of dense rocksalt and/or spinel-like surface phases[6,7]. For example, Boulineau et al. report the densification of layered materials as a result of transition metal segregation which originates from decoordination of transition metals on the surface[8]. Hence, target strategies to enhance oxygen retention such as surface doping or coatings are of interest. Coating procedures, either by atomic layer deposition or chemical processes, are known as an efficient procedure to protect surfaces from various degradation processes including hydrofluoric acid attack and metal dissolution[9,10]; however, surface coatings may also impede Li mobility[11]. On the other hand, it is reported that surface doping can minimize phase segregation or separation at interfaces during the cycling process[12]. For example, Ti and Al surface doping of layered $Li(Ni_{1/3}Co_{1/3}Mn_{1/3})O_2$ have shown enhanced discharge capacity at low temperature indicating facile ion transport[13,14]. Other efforts show that F surface doping provides improved cyclability[15] and that Co and Ti doping effectively prevents the Mn dissolution[12,16] in the $LiMn_2O_4$ spinel. A recent computational study also shows that surface doping with Y, Gd, La, and Zr can influence the particle morphology of $MnAl_2O_4$ spinel by selectively promoting surface facet stability[17]. In a recent study, Si, Ti, V, and Zr were considered as potential surface dopants to prevent transition metal dissolution in the high-voltage spinel $LiNi_{0.5}Mn_{1.5}O_4$[18]. However, despite pioneering studies on surface doping using both experiments[13,14,16] and computations[17], to the best of our knowledge, a broad investigation of surface dopant effect on oxygen evolution and surface protection has not yet been presented.

In this work, we comprehensively screen for optimal elements to act as surface dopants with the specific role of oxygen retention using the first principles Density Functional Theory (DFT). We systematically rank suitable dopants—to be added during synthesis—by investigating their preferential segregation to the surface as compared to the bulk of the material. In addition, we introduced the surface defect formation energy as our second criteria so as to minimize the risk of forming secondary impurity phases during synthesis. Finally, surface oxygen retention is investigated for the remaining candidates as compared to the surface of the original active material. All transition metals, post-transition metals, and metalloids are considered and we specifically target layered Li-excess, Mn-rich materials by using the end member $Li_2MnO_3$, as a model representative cathode system. Finally, one of the computationally identified top-candidate dopants was selected to demonstrate the potential value of the approach. A Ta-doped, Li-excess $Li_{1.3}Nb_{0.3}Mn_{0.4}O_2$ cathode material was synthesized and shown to demonstrate the enhanced presence of Ta at the surface of the material, initial improved electrochemical performance, and significantly improved oxygen retention, as compared to the un-doped material.

## Results

**Computational screening.** To identify the best candidate surface doping elements and elucidate the mechanism behind their effectiveness, we considered three major factors as selection criteria. First of all, a surface dopant element added during the synthesis process should preferentially occupy the surface region as compared to the bulk, hence we evaluate the dopant segregation energy ($E_S$), defined as

$$E_S = \Delta E^{\text{bulk}} - \Delta E^{\text{surface}} \tag{1}$$

to classify surface vs. bulk doping elements. Here, $\Delta E^{\text{bulk}}$ represents the energy difference between the dopant-containing bulk system and the pristine bulk system, and $\Delta E^{\text{surface}}$ represents the equivalent quantity for the surface. We assume the candidate dopant will occupy the octahedral transition metal site, which is certainly an approximation as some of the cations may prefer a Li site substitution. However, most of the site energy difference is expected to cancel between the doped surface and the doped bulk in the resulting $E_S$. Furthermore, the relative dopant segregation energy depends on the particular surface facet. For instance, Zn doping in the bulk is preferred compared to doping at the (001) surface but less preferred than doping on the (110) surface (see Fig. 1). Hence, the dopant segregation energy is examined for all stable, low-index surfaces of the system. In the low Miller index limit, layered $Li_2MnO_3$ exhibits five stable surfaces which contain two dominant surfaces ((001) and (010)) and three subsurfaces ((100), (110), and (111))[2]. The entire list of segregation energies, $E_S$, is presented in Fig. 1, for the considered transition metal, post-transition metal, and metalloid dopants.

A priori, we expect the dopant segregation energy ($E_S$) to be influenced, e.g., by dopant cation radius, coordination number, and degree of bonding between the dopant and its nearest neighbor environment. For example, if the ionic radius of the dopant is significantly different as compared to the host cation, such an element would preferentially segregate to the surface due to the strain penalty associated with the size difference between the host cation and dopant[19]. The ionic radii of the dopants were obtained by analyzing the resulting oxidation state of the dopant, either in the host matrix or at the surface, as obtained by the DFT calculations. We also investigated whether the oxidation state of the dopant is dependent on dopant concentration, which, in the limit of doping concentration (up to 50% of surface) was found constant. Given the oxidation state and coordination, the radii were extracted from tabulated Shannon radii[20]. Our investigation confirms that dopants with large ionic radii as compared to the host cation indeed segregate toward the surface which is shown by the correlation between the radius of the dopants ($R_{ion} > R_{Li}$; green to purple colors of the element labels in Fig. 1) and dopant segregation ($E_S > -0.026$ eV (estimated energy fluctuation at 300 K from Boltzmann distribution); light pink to blue colors of circular charts in Fig. 1). However, small differences—either larger or smaller—in ionic radius between the host cations and the dopants exhibit no strong preference. Indeed, due to the relatively small size of the host cation, this effectively limits the selection of preferred surface dopants to the larger cations. The ionic radius of the smallest dopant; $Si^{4+}$ ($R_{Si} = 0.4$ Å); is only 24% smaller than $Mn^{4+}$. From this first criteria in our screening strategy we can identify the elements that would—if added during synthesis—preferentially segregate to the surface of Li-excess layered $Li_2MnO_3$.

The second screening criteria is designed to eliminate elements which—while preferring the surface region over the bulk material—may exhibit such a high surface dopant formation energy that it becomes favorable to phase separate and create secondary

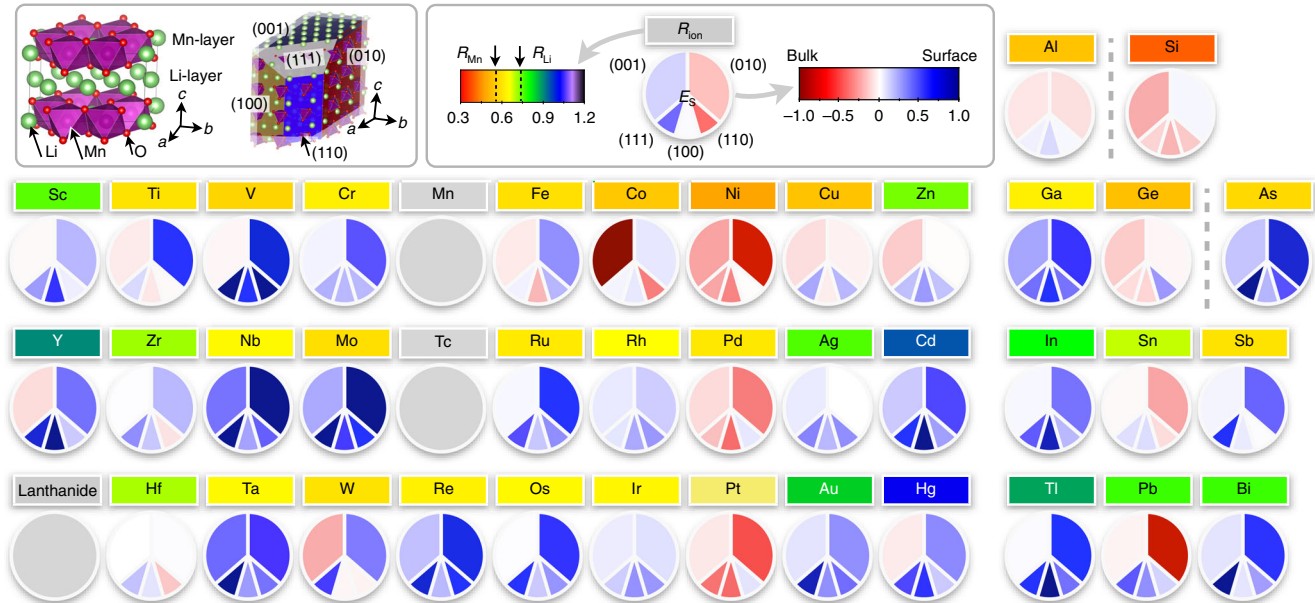

**Fig. 1** The dopant segregation energy as a function of dopant element for the 5 most stable low index surface facet in $Li_2MnO_3$. The circular charts are designed to give a visual overview of the surface segregation driving force by indicating red for bulk and blue for surface preference, respectively. The rainbow color scheme forming the background of the element label indicates the ionic radius of the dopant as analyzed using the result of the oxidation state of the dopant and tabulated Shannon radii. Additionally, the radii of the host cations $R_{Mn}$ (0.53 Å) and $R_{Li}$ (0.76 Å) are marked by dashed lines in the legend. A comprehensive list of dopant oxidation state and their ionic radius are presented in the supporting information (Supplementary Table 1)

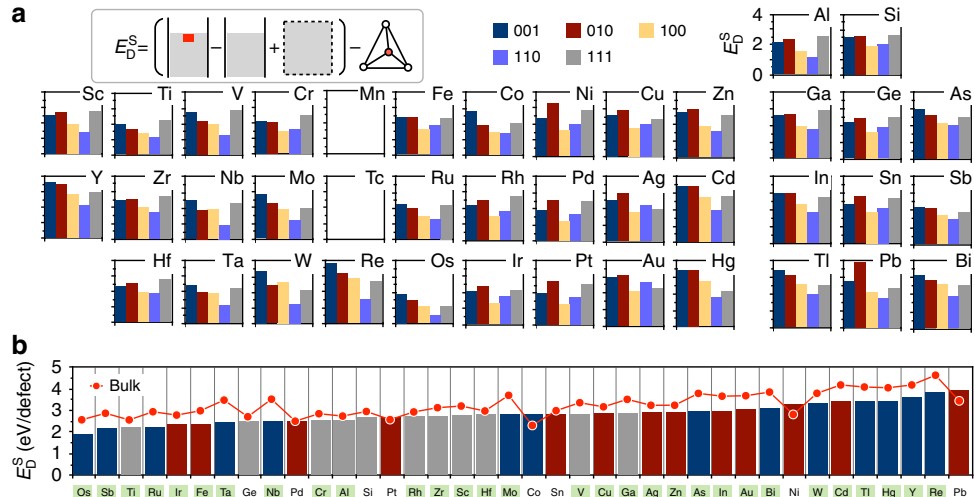

**Fig. 2** The defect formation energy for each stable surface facet. **a** The defect formation energies are varied in accordance with the facets. The inset schematic illustrates how the formation energy was obtained. **b** An ascending list of the surface doping formation energy of each dopant, where the green labels indicate elements which preferentially occupy the surface region, obtained from Fig. 1

impurity phases during synthesis. Hence, we examine the defect formation energy $\left(E_D^S\right)$ of the surface dopants defined as

$$E_D^S = \left(\Delta E^{surface} + E^{bulk}\right) - E_{PD}^{eq} \qquad (2)$$

Here, $\Delta E^{surface}$ is the energy difference between the surface with a defect and the pristine surface. To obtain a reasonable defect formation energy, we compare $\Delta E^{surface} + E^{bulk}$, which represents the doped surface formation energy, with the energy above the convex hull $(E_{PD}^{eq})$ of the respective stable, doped polymorphs. Here,

we used the $E_{PD}^{eq}$ values as listed in the Materials Project public open database[21]. We note that the approximation that all dopants substitute on a Mn site will in some cases result in a penalized defect formation energy which will push those candidates to higher $E_D^S$. The entire list of stable polymorphs for a given defect (dopant) element is presented in the supporting information (Supplementary Table 2). Figure 2a shows $E_D^S$ for all five stable surfaces throughout the considered cation dopants. Since all surface facets in layered $Li_2MnO_3$ are prone to oxygen loss as Li is extracted, we require an optimal dopant to occupy all surface facets in order to improve the performance of the material. To screen out dopants with

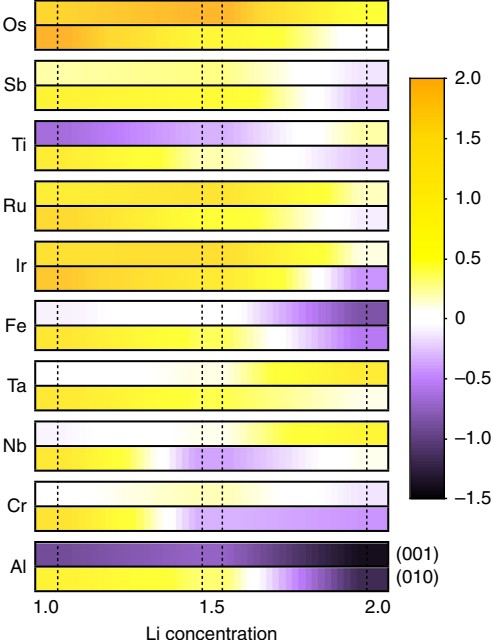

**Fig. 3** The relative surface oxygen release energies. The top 10 candidate dopants contained surfaces compared to the pristine systems shown for representative (001) and (010) surface facets. A dark yellow color indicates stronger oxygen retention, while a purple color indicates less protection against oxygen release as compared to the pristine, undoped surface

prohibitory large defect formation energy, we plot the maximum defect formation energy across facets as a function of the element. Figure 2b illustrates the maximum $E_{\mathrm{D}}^{\mathrm{S}}$ in ascending order for each dopant. To classify appropriate surface dopants, the candidate surface doping materials should satisfy $E_{\mathrm{S}} > 0$ and exhibit a reasonably small $E_{\mathrm{D}}^{\mathrm{S}}$ for stability.

Green colored-labeled elements in Fig. 2b indicate $\bar{E}_{\mathrm{S}} > -0.026$ eV, highlighting the elements which performed well under the surface preference criteria. Here, the average segregation energy $(\bar{E}_{\mathrm{S}})$ is calculated using a weighted average of the relevant surface area within the Wulff shape.

Combining the information from Figs. 1 and 2, we find that elements which preferentially occupy the surface region of Li-excess layered $Li_2MnO_3$, where the defect formation energy is still modest, are Os, Sb, Ti, Ru, Ir, Fe, Ta, Nb, Cr, and Al.

In the final screening, we examine the selected top 10 candidate dopants for improved oxygen retention at the surface. The oxygen evolution energy $(\tilde{E}_{\mathrm{O}})$ for a given surface facet is defined as[2]

$$\tilde{E}_{\mathrm{O}} = E_{\mathrm{O}-x'}^{\mathrm{slab}} + \Delta\mu_{\mathrm{O}} - E^{\mathrm{slab}} \qquad (3)$$

Here, $E_{\mathrm{O}-x'}^{\mathrm{slab}}$ is the oxygen-deficient slab energy, where $O-x'$ indicates the slab oxygen stoichiometry, $\Delta\mu$ is the oxygen chemical potential, and $E^{\mathrm{slab}}$ is the pristine slab energy. The oxygen chemical potential is defined as $\Delta\mu_{\mathrm{O}} = \mu_{\mathrm{O}} - \mu^{\star}$ where the reference chemical potential $(\mu^{\star})$ is obtained by calibrating the formation enthalpies with experimental measurements of various main group binary oxides[22,23]. To indicate the presence of a surface dopant, we define $\tilde{E}_{\mathrm{O}}$ of the un-doped (pristine) and doped systems as $\tilde{E}_{\mathrm{O}}^{\mathrm{pristine}}$ and $\tilde{E}_{\mathrm{O}}^{\mathrm{doped}}$, respectively. Consequently, the oxygen evolution energy difference, $\Delta\tilde{E}_{\mathrm{O}} = \tilde{E}_{\mathrm{O}}^{\mathrm{doped}} - \tilde{E}_{\mathrm{O}}^{\mathrm{pristine}}$, provides a measure of the oxygen retention capability for a given, doped surface.

Figure 3 shows $\Delta\tilde{E}_{\mathrm{O}}$ for the top 10 candidate dopants obtained from the two first selection criteria. Here, the oxygen evolution energies are reported as the average value of all symmetry-unique oxygen vacancies on each surface for a given defect. Each color in Fig. 3 represents the relative oxygen evolution energy $(\Delta\tilde{E}_{\mathrm{O}})$ as compared to the pristine surface, therefore, yellow (purple) colors indicate an improved (decreased) oxygen retention relative to the pristine surface.

Using Fig. 3, we conclude that, on average, Ti and Al dopants are not expected to enhance oxygen retention, while Os, Sb, Ru, Ir, and Ta are indeed predicted to improve it. Prior experimental efforts support these claims. For instance, Sathiya et al. reported a decreasing oxygen vacancy per unit formula by increasing Ru concentration in the solid solution of $Li_2RuO_3$ and $Li_2MnO_3$[24], where Ru is found to aggregate preferentially at the surface[25]. Hence, these results directly support that (1) the Ru dopant preferentially occupy the surface, and (2) the Ru doping prevents surface oxygen evolution. Conversely, Al doping of $Li_2MnO_3$[26], was actually found to enhance degradation and structural transformation as compared to the original material[27].

The underlying atomistic reason for the improved oxygen retention can be understood by elucidating the influence of the dopant on the local electronic structure of a specific surface facet. As an example, the highest and lowest ranked dopants within the top 10 candidates are Os and Al. Figure 4a shows the geometry of an Os dopant residing on the (001) surface and the corresponding electronic density of states (DOS) of the valence energy band (Fig. 4b). Here, red solid lines represent the DOS for Os $p$-orbitals and black dashed lines denote the DOS of neighboring oxygen $p$-orbitals. The results show strong hybridization which evidences the increased charge sharing between the Os and its neighboring O ions (see blue highlight in Fig. 4c). On the other hand, Al doping of the (001) surface (Fig. 4d) presents weak hybridization of DOS between Al $p$-orbitals (yellow solid lines in Fig. 4e) and O $p$-orbitals (black dashed lines in Fig. 4e). Reduced charge sharing between Al and its neighboring O ions is shown as compared to the original, undoped material (Fig. 4f). We emphasize that there may be other benefits associated with Al doping, however, oxygen retention on Mn-rich, Li-rich surfaces is not supported by the present results.

Interestingly, we find that the strong oxygen binding elements exhibit a positive effect on all surface facets, e.g., the effect is universal irrespective of surface structure as well as Li content. Hence, it is reasonable to find the root cause in the simple binding between a metal cation and oxygen. Oxygen binding to transition metals has been widely studied in the context of chemisorption or adatom binding energies on metal surfaces[28–31]. These studies pioneered and showcased the $d$-band center model, such that high oxygen binding energy increases with the period within the same group. For example, Ir has higher binding energy than Rh and Co, and Os has higher energy than Ru and Fe. Among the transition metal elements with high oxygen binding energy are Au, Ag, Pt, Rh, Ir, Cu, Co, and Ni and Pd[28,29] whereof—in our filtering—only Ir and Ru remain after the screening on surface preference and defect formation energy.

**Experimental verification**. The computational screening procedure highlighted Os, Sb, Ru, Ir, and Ta as optimal elements for oxygen-retaining surface dopants in Li-excess Mn-rich cathodes. Accordingly, one of these dopant elements was chosen, Ta, to validate and demonstrate the value of the theoretical predictions by experimental synthesis, electrochemical testing, and characterization. A pristine $Li_{1.3}Nb_{0.3}Mn_{0.4}O_2$ (LNMO) within the Fm-3m space group (lattice constant 4.195 Å) was synthesized by a molten-salt method as a reference, undoped system. The doped

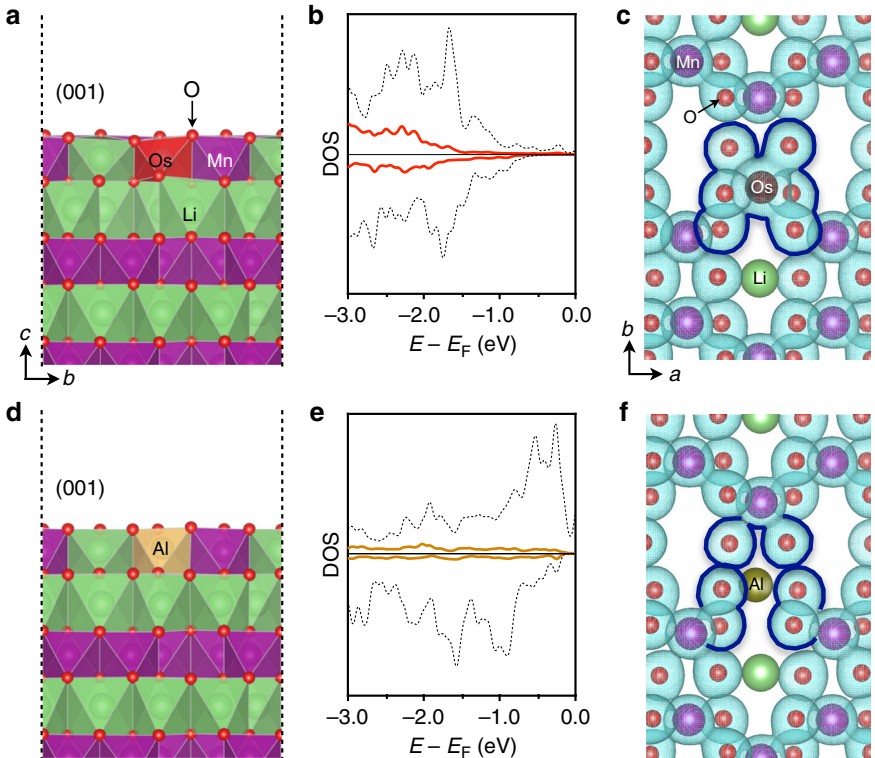

**Fig. 4** Electronic structure of surface Os and Al dopants, highlighting their bonding to neighboring oxygen ions. The Os doping on the (001) surface (**a**) presents strong hybridization between oxygen $p$-orbitals (black dashed line in **b**) and the surface defect $d$-orbitals (red solid lines in **b**). The hybridization is further illustrated by the charge (cyan isosurface lines, here iso-surface levels of **c** and **f** are 0.1 Å$^{-3/2}$) sharing between the surface defect atom and neighboring oxygen atoms (**c**). In contrast, Al doping on the (001) surface (**d**) exhibits weak hybridization (**e**), which is evidenced in charge separation between the defect atom and its neighboring oxygen ions (**f**)

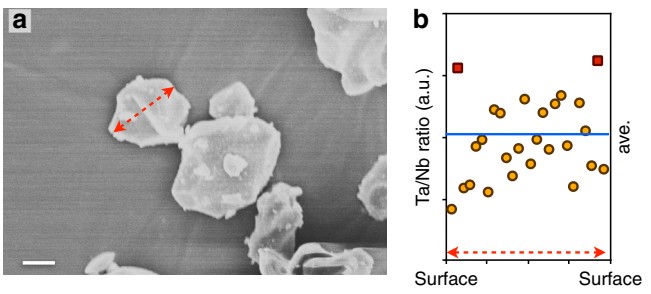

**Fig. 5** Ta-doped LNMO and the Ta distribution in the particle. SEM image (scale bar, 2 μm) of the as-prepared $Li_{1.3}Nb_{0.285}Ta_{0.015}Mn_{0.4}O_2$ (**a**), and the energy dispersive X-ray spectroscopy (EDX) line scan (**b**)

system was prepared by solid state synthesis (see Methods) with an average particle size of ca. $3 \times 10^{-6}$ to $10 \times 10^{-6}$ m and was found to incorporate 2% Ta, as determined by ICP. Phase purity was confirmed by both X-ray and neutron diffraction studies, indicating that the doped sample has a chemical formula of $Li_{1.3}Nb_{0.285}Ta_{0.015}Mn_{0.4}O_2$ (Supplementary Figure 1). Energy dispersive X-ray spectroscopy (EDX) was used to track the dopant distribution within the LNMO particle. Figure 5a shows the SEM image of 2% Ta-doped LNMO and the respective EDX line scan (Fig. 5b). Figure 5b shows that the Ta/Nb ratio at the edge of Ta-doped LNMO particle (red square) is indeed relatively higher than the center part of the particle (yellow circle). Hence, the results support the computed prediction that Ta tends to segregate towards the surface, however, we also note a significant

fluctuation in the Ta/Nb ratio due to the rough particle morphology. Given an average dopant concentration of 2%, we estimate approximately 4% Ta dopant concentration at the surface.

The electrochemical performance of the doped and undoped LNMO was evaluated in both galvanostatic and cyclic voltammetric modes. For the galvanostatic mode, Ta-doped LNMO cells were cycled at 10 mA g$^{-1}$ between 1.5 and 4.8 V. As shown in Fig. 6a, the Ta-doped LNMO exhibits a slightly higher specific capacity for both charging and discharging cycles. The cyclic voltammograms (CV) between the undoped and the Ta-doped LNMO present similar behaviors (Fig. 6b–d), indicating consistent bulk reaction mechanisms for Mn and O ions. We note that the initial specific capacity of Ta-doped LNMO (Fig. 6d) is higher than the undoped LNMO (Fig. 6c) for both charging and discharging cycles, signifying a beneficial effect, despite low amounts of Ta doping. However, as the cells were cycled, voltage fading and capacity reduction were observed for both samples. After 10 cycles, the specific capacity of the Ta-doped sample is marginally higher than that of the undoped for both charging and discharging cycles (Supplementary Figure 2).

Finally, the differential electrochemical mass spectroscopy (DEMS) measurements (Supplementary Figure 3) clearly demonstrate less oxygen evolution for the Ta-doped LNMO (12 μmol g$^{-1}$) as compared to the pristine LNMO (41 μmol g$^{-1}$). Here, $CO_2$ evolution also occurs from each material and is consistent with gas evolution from other transition metal oxides where $CO_2$ was observed to evolve predominantly from the oxidation of residual impurities (e.g., $Li_2CO_3$) at potentials lower than 4.8 V[32]. A potential explanation for the greater $CO_2$ evolution from the Ta-doped sample is that a slightly greater excess of $Li_2CO_3$ is used

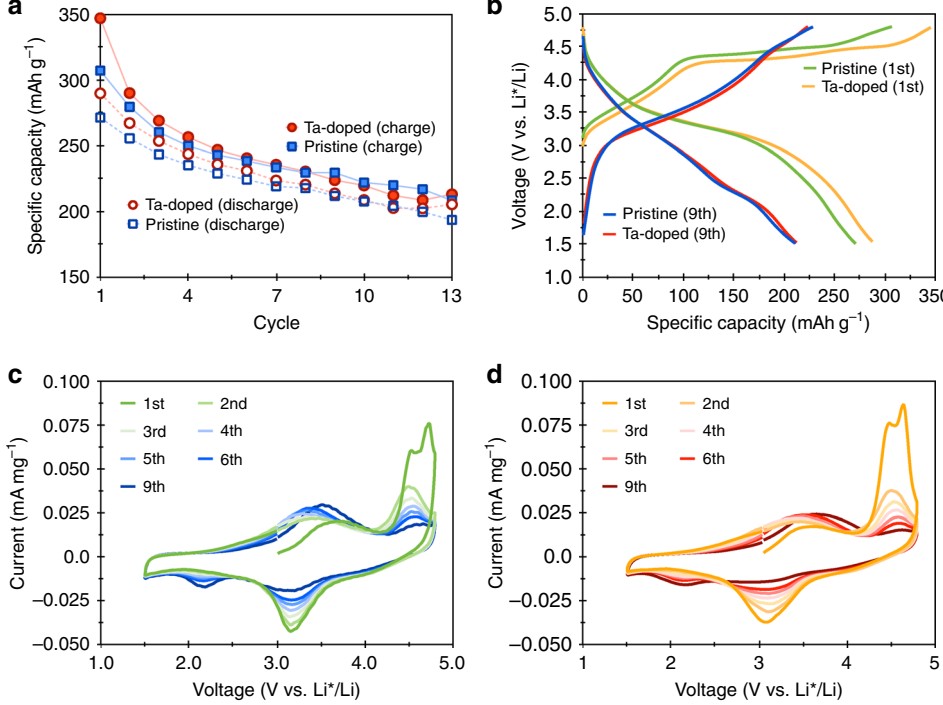

**Fig. 6** Cycling performance of pristine and Ta-doped LNMO. The specific capacity of Ta-doped and pristine LNMO over the first 13 cycles (**a**). The Ta-doped samples exhibit higher initial specific capacity compared to the pristine samples. The first and ninth full charge and discharge curves of undoped (green and blue lines) and Ta-doped (yellow and red lines) LNMO (**b**). The cyclic voltammograms of the first 9 cycles for the pristine LNMO (**c**) and Ta-doped LNMO (**d**)

for, and remains in residual quantities after, its synthesis compared to the undoped sample. Residual $Li_2CO_3$ was confirmed with simple titrations indicating 0.85 wt% $Li_2CO_3$ in the pristine $Li_{1.3}Nb_{0.285}Ta_{0.015}Mn_{0.4}O_2$ and 5.74 wt% in the Ta-doped material. If all $Li_2CO_3$ in each sample were completely oxidized to evolve $CO_2$, this oxidation process would account for 20% (pristine) and 37% (Ta-doped) of the total $CO_2$ evolved during the first delithiation of each respective material, implying that electrolyte degradation accounts for a portion of $CO_2$ evolved. Further studies are necessary to fully understand the origin of $CO_2$ evolution for these materials, as the presence of $Li_2CO_3$ can clearly affect the stability of the electrolyte at high voltages[33]. Nevertheless, our DEMS results clearly confirm that Ta-doping effectively reduces the evolution of oxygen from the oxide lattice.

## Discussion

We applied a three-tier high-throughput computational screening of possible dopants, spanning all transition metals, post-transition metals, and metalloids in order to guide the selection of the best candidate cation dopant to enhance surface oxygen retention in Li-rich, Mn-rich Li-ion cathodes. First-principles DFT was employed to systematically evaluate elements as potential surface dopants to enhance oxygen retention. A three-tier hierarchical screening process was employed to reflect: (i) the dopant segregation energy considering each stable surface facet, (ii) the thermodynamic stability of the surface defect formation energy, and (iii) the thermodynamic driving force for surface oxygen retention. We note that all cation substitutions were assumed on the transition metal site. This approximation will penalize the defect formation energy of some cations that would preferentially occupy the Li sites, but we expect the site energy difference to mostly cancel in the

segregation energy. In the first tier, we found that large dopants showed a strong correlation with desirable surface segregation. In the second tier, the 10 cations with the smallest surface defect formation energy were chosen and subjected to tier 3, where their surface bonding to oxygen was examined and compared to the pristine material. Interestingly, we found that cations which exhibit strong hybridization with its surrounding oxygen significantly enhanced oxygen retention. In contrast, cations such as $Al^{3+}$ and $Ti^{4+}$ where the charge sharing with oxygen was found to be smaller, exhibited weaker bonds to oxygen. Furthermore, results from tier 3 were found to be largely independent of surface morphology, which indicate that the effect is inherent in the local chemical bonding rather than structure. Hence, we would expect these results to be transferable to the broader family of Li-ion cathodes. Finally, top candidates were identified as Os, Sb, Ru, Ir, and Ta, whereof Ta was chosen to validate the predictions using solid state synthesis of Ta-doped $Li_{1.3}Nb_{0.3}Mn_{0.4}O_2$, subsequent electrochemical testing, and characterization of gas release and surface speciation. An overall doping level of 2% was achieved with an enrichment of Ta at the surface corresponding to approximately 4%, validating the surface segregation preference of Ta. Despite the low amounts of Ta, a modest improvement in electrochemical performance and, a significantly improved oxygen retention was demonstrated.

## Methods

**Computations**. The total energy results were calculated using the first principles DFT which is utilized by Vienna ab initio Simulation Package (VASP)[34–37] with the projector augmented wave (PAW)[38,39] pseudopotential method. The exchange correlation functional is chosen as the generalized gradient approximation (GGA +U)[40–42] with an on-site Hubbard parameter ($U_{Mn}$ = 3.9 eV[43]). The calculations were converged within 1 meV, enabled by a cutoff energy of 520 eV, and k-point sampling density of 1000 (k-points per reciprocal cell), adjusted by the size of the supercell.

**Synthesis**. For a typical molten salt synthesis of pristine $Li_{1.3}Nb_{0.3}Mn_{0.4}O_2$, stoichiometric of ACS-graded (99%+) $Li_2CO_3$, $Nb_2O_5$, $Mn_2O_3$ precursors were ball-milled in a RETSCH Planetary Ball Mill PM 100 at 200 rpm for 12 h, using zirconia balls/jar and ethanol as solvent. To compensate the loss of lithium at elevated temperature, 10% extra lithium salt was added during the milling process. The dried powder was further mixed with potassium chloride with different molar ratio ($R$ = mole of KCl/total mole of transition metal precursors; $2.5 \leq R \leq 5$). The mixture was heated at 950 °C under Ar atmosphere for 12 h with a ramp rate of 4 °C/min for both heating and cooling. After the reaction, KCl was dissolved in water and the final product was obtained by a simple filtration. Ta-doped sample $Li_{1.3}Nb_{0.285}Ta_{0.015}Mn_{0.4}O_2$ was prepared by conventional solid-state method. Stoichiometric of ACS-graded (99%+) $Li_2CO_3$, $Nb_2O_5$, $Mn_2O_3$ precursors were ball-milled in a RETSCH Planetary Ball Mill PM 100 at 200 rpm for 12 h, using zirconia balls/jar and ethanol as solvent. To compensate the loss of lithium at elevated temperature, 15% extra lithium salt was added during the milling process. The precursor was heated at 950 °C under Ar atmosphere for 12 h with a ramp rate of 4 °C min$^{-1}$ for both heating and cooling.

**Characterization**. Phase purity was first analyzed by a Bruker D2 powder X-ray diffractometer (Cu Kα, 40 kV, 30 mA). The diffraction data was analyzed by GSAS/EXPGUI packet. Scanning electron microscopy (SEM) images were obtained in a JOEL JSM-7610F Scanning Electron Microscope (10 kV, 10 mA) using secondary electron mode. For electrochemical measurements, the samples were investigated in 2032-type coin cells with half-cell configuration. The active material was first ball-milled with carbon black (20 wt%) to reduce the particle size and improve its electronic conductivity. The electrode slurry was prepared by the milled product (80 wt%), carbon black (10 wt%) and polyvinylidene fluoride (PVDF) binder (10 wt%) using N-methyl-2-pyrrolidone (NMP) as solvent. The electrodes were dried in a vacuum oven at 120 °C for 12 h. A 1 M lithium hexafluorophosphate ($LiPF_6$) solution dissolved in ethylene carbonate and diethyl carbonate (EC/DEC = 50/50 (v/v)) was used as electrolyte for all investigated samples. The samples were studied in both galvanostatic and cyclic voltammetric modes using Bio-Logic VMP3 Multi-channel workstation. For the DEMS measurement, the electrode was prepared in a similar fashion as outlined above, but instead used an aluminum mesh current collector to allow gases to be collected on the back side of the electrode. A customized Swagelok-type cell that ensured hermetic integrity was assembled using a Li metal anode and the aluminum mesh supported cathode in an Ar glovebox. The cell was connected via capillaries to a gas handling unit that allowed 500 μl pulses of a carrier gas to be swept through the cell head space and sent to a mass spectrometer for partial pressure analysis, thereby allowing quantitative analysis of the gaseous products[32,44]. A 1 M lithium hexafluorophosphate ($LiPF_6$) solution dissolved in ethylene carbonate and diethyl carbonate (EC/DEC = 50/50 (v/v)) was used as electrolyte for all investigated samples. The samples were studied in galvanostatic mode using a BioLogic SP-300 potentiostat at a constant current rate of 25 mA g$^{-1}$.

## Data availability

The data used in this work is available at the Materials Project (http://materialsproject.org).

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

## Acknowledgements

This work was intellectually led by the Battery Materials Research (BMR) program under the Assistant Secretary for Energy Efficiency and Renewable Energy in the Office of Vehicle Technologies of the U.S. Department of Energy, Contract No. DE-AC02-05CH11231. Computational resources were provided by the National Energy Research Scientific Computing Center (NERSC). The Stanford Synchrotron Radiation Lightsource, SLAC National Accelerator Laboratory, were supported by the U.S. Department of Energy, Office of Science, Office of Basic Energy Sciences under Contract No. DE-AC02-76SF00515. We are grateful to Lori Kaufman for guidance on the carbonate titrations.

## Author contributions

Y.S. and K.A.P. developed the project. Y.S., M.A., and K.A.P. calculated the high-throughput DFT computation. W.H.K. and G.C. synthesized, and electro-chemical measurement of the Ta-doped LNMO. W.H.K., J.K.P., G.C., and B.D.M. characterized the active materials. All authors contributed to the writing of the manuscript.

## Additional information

**Competing interests:** The authors declare no competing interests.

