## [Peer Review File · Nature Communications]

Reviewers' comments:

Reviewer #1 (Remarks to the Author):

The authors have done a very nice study on the possibilities of changing surface structures through dopants in order to mitigate the effects of oxygen evolution in Li-rich materials. These results will certainly be of interest to the community. The authors present their results clearly and discuss the relation to experiment by example. The results of the experiment are presented in a fair manner with discussion of positive results and well as neutral and/or negative results and I think many experimental groups will be interested in following up.

It is interesting to me that the doped sample appears to have lower first-cycle efficiency, even though less oxygen is evolved. If the authors have any comments on this observation they would be interesting to hear. Is it possible that higher impurities cause this, or, is more CO₂ being evolved from lattice oxygen in the doped sample for some reason? It is reasonable to expect that CO₂ may also originate from evolved lattice oxygen and subsequent reaction as has been previously reported.

The authors mention the possible effects of Li₂CO₃. The Ta doped sample does use 15% excess in the synthesis as compared to 10% excess in the undoped sample. I'm not sure XRD is the best way to quantify, perhaps something like XPS may be more useful.

Might be better to mass normalize(cathode active)6c and 6d as well as S4.

Please check the sample labeling of figure S2.

Reviewer #2 (Remarks to the Author):

This is an interesting and comprehensive study (computationally at least) on the possibility of using surface coating to mitigate the effects of oxygen evolution in Li-rich cathode materials. The computational results are compelling and generally well presented, though I would prefer the entries for Tc in figs. 1 and 2 to be left blank!

For the experimental work note that D in DEMS refers to differential (correct in supplementary). The DEMS results must be presented more clearly, particularly given the large amount of CO₂ released by the Ta containing material. The actual amount of O₂ produced is the important parameter and this must be made clearer. Overall the reduction in O₂ release and improvement in cycling performance are not very convincing. Ideally the authors should provide more compelling experimental evidence with greater surface enrichment.

Lastly a higher standard of proof reading is needed.

Reviewer #3 (Remarks to the Author):

This is a combined high-throughput computation study for identifying dopants for Li₂MnO₃ cathode material to suppress oxygen evolution, including some experimental support.

The computation is novel as no similar surface-based high-throughput computation is done before. In addition, the computation carefully considered different surfaces, dopant segregation energy versus defect formation energy. Some technical problems of computation should be clarified:

- In the calculations of dopant segregation and defect formation, is the effect of different Li concentration considered? How about the configuration of dopant?

Such consideration may be important as the dopant segregation may change during repeated

cycling, affecting its effect.

- For oxygen defect, did the calculations consider different site of oxygen vacancy considered ?
- Fig. 4 should be explained better. What are the lines of different color in the DOS? It is not clear why charge density plots are the indication of the hybridization, as the choice of different isosurface would affect the plots.
- Also would surface reconstruction (especially after cycling) have an effect ?

For the experiments part, the positive effect of Ta is very small. While O₂ evolution seems to be suppressed, the overall gas evolution is worse. Is this because the enhanced catalytical effect of the dopant? This suggests that the proposed doping is not a useful and practical strategy and has limited impact.

Reviewer #1

We are very gratified for the reviewer positive comments. We have carefully considered each of the questions posed by the referee (see below) and made changes where appropriate.

>> It is interesting to me that the doped sample appears to have lower first-cycle efficiency, even though less oxygen is evolved. If the authors have any comments on this observation they would be interesting to hear.

Good question and warrants further investigation. Compared to the un-doped material, surface Ta-doping significantly improves oxygen retention as demonstrated by the DEMS measurements. However, it is possible that doping also leads to changes in the interaction and side reactions with the liquid electrolyte which may be responsible for the reduced efficiency, especially on the fresh surface in the first cycle. For example, the increased amount of Li_2CO_3 used in the Ta-doped sample as compared to the pristine is one difference that may contribute, as commented upon below. However, we emphasize that the observed better long-term stability in the doped sample demonstrates the importance of oxygen retention and structural stabilization in the oxide performance.

>> Is it possible that higher impurities cause this, or, is more CO_2 being evolved from lattice oxygen in the doped sample for some reason? It is reasonable to expect that CO_2 may also originate from evolved lattice oxygen and subsequent reaction as has been previously reported.

Indeed, the DEMS measurements demonstrate less oxygen evolution for the Ta-doped as compared to the pristine sample. CO_2 evolution also occurs from each material and is consistent with gas evolution from other transition metal oxides where CO_2 was observed to evolve *predominantly* from the oxidation of residual impurities (e.g. Li_2CO_3) at potentials lower than 4.8 V. (see Renfrew, S. E. and McCloskey, B. D. Journal of the American Chemical Society 2017, 139, 17853–17860, PMID: 29112815) Hence, we believe that the greater CO_2 evolution from the Ta-doped sample is the result from a greater excess of Li_2CO_3 used for, and remains in residual quantities after, its synthesis compared to the undoped sample. To further address this question, we added carbonate titration results that confirm that Li_2CO_3 persists to a greater extent in the Ta-doped sample

compared to the undoped sample. However, we note for these specific transition metal oxide compositions, electrolyte degradation cannot be fully ruled out as a possible contributor to CO₂ evolution. These results, as well as other recent results (see Mahne et al., *Angew. Chem., Int. Ed.* 2018, 57, 5529), strongly suggest a correlation between Li₂CO₃ content and electrolyte stability at high voltages. Further studies are necessary to fully understand the CO₂ evolution for these materials, but clearly Ta-doping effectively reduces the evolution of lattice oxygen. We have incorporated language to this effect in the paper on page 13.

The authors mention the possible effects of Li₂CO₃. The Ta doped sample does use 15% excess in the synthesis as compared to 10% excess in the undoped sample. I'm not sure XRD is the best way to quantify, perhaps something like XPS may be more useful. Indeed, the two samples were prepared with a slightly different content of Li₂CO₃ in the synthesis as those were the optimized conditions to get phase-pure samples, as confirmed by lab PXRD. When the same amount of Li₂CO₃ (10%) was used to prepare Ta doped, a small amount of Li₃NbO₄ was always detected. In the attached figure (see ppt slide #1), we show the O K-edge sXAS spectra collected on both samples (BL 10-1 at SSRL). A comparison to the standard O K-edge spectra of Li₂CO₃ (ref. Qiao et al., *PLOS ONE* 7 (2012) e4918) suggests that in both samples, the presence of residue Li₂CO₃ is relatively small. Unfortunately, XAS is relatively insensitive to detect differences between the samples in residual quantities of Li₂CO₃ to the accuracy needed. Therefore, additional Li₂CO₃ titrations were performed to quantify the amount of Li₂CO₃ in each material, and show that substantially more Li₂CO₃ was indeed present in the Ta-doped sample than the undoped sample. We have included the relative wt% of residual Li₂CO₃ in each material (0.85 wt% for the undoped sample, 5.74 wt% for the Ta-doped sample) in the updated manuscript, as well as updated the discussion on the influence of the presence of Li₂CO₃ on our gas evolution results.

Might be better to mass normalize(cathode active)6c and 6d as well as S4.
Thank you for the suggestion. Figure 6 and S4 have been updated.

Please check the sample labeling of figure S2.
Thank you ; the figure S2 caption has been updated.

Reviewer #2

We thank the referee's positive review and useful feedback. We have made several changes based on the specific comments as outlined below.

The computational results are compelling and generally well presented, though I would prefer the entries for Tc in figs. 1 and 2 to be left blank!

We updated Tc parts in the Figure 1 and 2.

For the experimental work note that D in DEMS refers to differential (correct in supplementary).

This is corrected.

The DEMS results must be presented more clearly, particularly given the large amount of CO₂ released by the Ta containing material. The actual amount of O₂ produced is the important parameter and this must be made clearer. Overall the reduction in O₂ release and improvement in cycling performance are not very convincing. Ideally the authors should provide more compelling experimental evidence with greater surface enrichment.

Thank you for the suggestions. The DEMS result has been updated in both the manuscript and the SI.

Lastly a higher standard of proof reading is needed.

We have carefully reviewed our manuscript.

Reviewer #3

We thank the referee for the positive review. We have made several changes based on the specific comments as outlined below.

- In the calculations of dopant segregation and defect formation, is the effect of different Li concentration considered? How about the configuration of dopant? Such consideration may be important as the dopant segregation may change during repeated cycling, affecting its effect.

Excellent questions. First of all, we investigated all possible dopant sites (Li sites; 2c, 2b, 4h / Mn site; 4g) on the surface, and selected the most stable site as a reference. According to our previous work (Adv. Energy Mater. 4 (2014) 1400498), the Mn migration is enhanced by Li-extraction (e.g. formation of Li vacancies). Whether the dopants would migrate – similar to the Mn – is an interesting question and requires the calculation of Li-concentration dependent activation barriers for each dopant and hence a serious paper in itself. However, we will venture a guess. The final dopant candidates: Ir⁵⁺, Ru⁵⁺, Os⁵⁺, Sb⁵⁺, and Ta⁵⁺ all show very strong octahedral preference, with high oxidation states. Since the migration to the Li layer usually occurs through an intermediate tetrahedral site, and high valence increases the activation barrier by repulsion to the nearest cation – we suspect that migration of these dopants into the Li layer would be very unfavorable, to the point of negligible.

- For oxygen defect, did the calculations consider different site of oxygen vacancy considered?

Yes, we considered the oxygen vacancies of all possible oxygen sites on each surface and present the average values - as a function of Li concentration - for each surface. For example, we calculated 6 vacancy sites on the 001 surface,

and 8 sites on the 111 surface. For clarification, we added a sentence on page 8 as, “Here, the oxygen evolution energies are reported as the average value of all symmetry-unique oxygen vacancies on each surface for a given defect.”

- Fig. 4 should be explained better. What are the lines of different color in the DOS? It is not clear why charge density plots are the indication of the hybridization, as the choice of different isosurface would affect the plots.

We apologize for any confusion. The color difference in the DOS is explained in the figure caption and the manuscript (page 9). The hybridization would satisfy two conditions which are 1) two ions have similar electron energy eigenvalues and 2) wave function overlap. The DOS plots of Figure 4 present the energy eigenvalue similarity, and the charge plots (as representing the square of the wave functions) present wave function overlap. Therefore, we would advocate that the DOS and charge plot provide a qualitative description of hybridization. Indeed, we note that the charge overlap between Os-oxygen and Al-oxygen exhibit marked differences for a broad range of isosurface values. This is illustrated in the figures below, which present the charge sharing with 3 different iso-surface levels (i.e., 0.4, 0.1, 0.04 Å^{-3/2}). In summary, we believe there is a strong correlation between the dopant oxygen binding energy, hybridization of the electron densities and – ultimately – oxygen retention in the cathode. For clarification, we specified the actual iso-surface value in the figure caption of the figure 4 on page 11.

- Also would surface reconstruction (especially after cycling) have an effect? For the experiments part, the positive effect of Ta is very small. While O2 evolution seems to be suppressed, the overall gas evolution is worse. Is this because the enhanced catalytical effect of the dopant? This suggests that the proposed doping is not a useful and practical strategy and has limited impact.

Good points and similar to that of the other reviewers. We kindly refer to the updated DEMS and titration result in the manuscript and the SI as well as previous answers to the reviewers.